# Epithelial Cell Adhesion Molecule (EpCAM) Expression in Human Tumors: A Comparison with Pan-Cytokeratin and TROP2 in 14,832 Tumors

**DOI:** 10.3390/diagnostics14101044

**Published:** 2024-05-17

**Authors:** Anne Menz, Nora Lony, Maximilian Lennartz, Sebastian Dwertmann Rico, Ria Schlichter, Simon Kind, Viktor Reiswich, Florian Viehweger, David Dum, Andreas M. Luebke, Martina Kluth, Natalia Gorbokon, Claudia Hube-Magg, Christian Bernreuther, Ronald Simon, Till S. Clauditz, Guido Sauter, Andrea Hinsch, Frank Jacobsen, Andreas H. Marx, Stefan Steurer, Sarah Minner, Eike Burandt, Till Krech, Patrick Lebok, Sören Weidemann

**Affiliations:** 1Institute of Pathology, University Medical Center Hamburg-Eppendorf, 20246 Hamburg, Germany; a.menz@uke.de (A.M.); nora_lony@hotmail.de (N.L.); m.lennartz@uke.de (M.L.); s.dwertmann-rico@uke.de (S.D.R.); r.schlichter@uke.de (R.S.); s.kind@uke.de (S.K.); v.reiswich@uke.de (V.R.); f.viehweger@uke.de (F.V.); d.dum@uke.de (D.D.); luebke@uke.de (A.M.L.); m.kluth@uke.de (M.K.); n.gorbokon@uke.de (N.G.); c.hube@uke.de (C.H.-M.); c.bernreuther@uke.de (C.B.); t.clauditz@uke.de (T.S.C.); g.sauter@uke.de (G.S.); a.hinsch@uke.de (A.H.); f.jacobsen@uke.de (F.J.); s.steurer@uke.de (S.S.); s.minner@uke.de (S.M.); e.burandt@uke.de (E.B.); t.krech@uke.de (T.K.); p.lebok@uke.de (P.L.); s.weidemann@uke.de (S.W.); 2Department of Pathology, Academic Hospital Fuerth, 90766 Fuerth, Germany; andreas.marx@klinikum-fuerth.de; 3Institute of Pathology, Clinical Center Osnabrueck, 49076 Osnabrueck, Germany

**Keywords:** EpCAM, TROP2, CKpan, tissue microarray, immunohistochemistry

## Abstract

EpCAM is expressed in many epithelial tumors and is used for the distinction of malignant mesotheliomas from adenocarcinomas and as a surrogate pan-epithelial marker. A tissue microarray containing 14,832 samples from 120 different tumor categories was analyzed by immunohistochemistry. EpCAM staining was compared with TROP2 and CKpan. EpCAM staining was detectable in 99 tumor categories. Among 78 epithelial tumor types, the EpCAM positivity rate was ≥90% in 60 categories—including adenocarcinomas, neuroendocrine neoplasms, and germ cell tumors. EpCAM staining was the lowest in hepatocellular carcinomas, adrenocortical tumors, renal cell neoplasms, and in poorly differentiated carcinomas. A comparison of EpCAM and CKpan staining identified a high concordance but EpCAM was higher in testicular seminomas and neuroendocrine neoplasms and CKpan in hepatocellular carcinomas, mesotheliomas, and poorly differentiated non-neuroendocrine tumors. A comparison of EpCAM and TROP2 revealed a higher rate of TROP2 positivity in squamous cell carcinomas and lower rates in many gastrointestinal adenocarcinomas, testicular germ cell tumors, neuroendocrine neoplasms, and renal cell tumors. These data confirm EpCAM as a surrogate epithelial marker for adenocarcinomas and its diagnostic utility for the distinction of malignant mesotheliomas. In comparison to CKpan and TROP2 antibodies, EpCAM staining is particularly common in seminomas and in neuroendocrine neoplasms.

## 1. Introduction

Epithelial cell adhesion molecule (EpCAM) is a type I transmembrane glycoprotein which was initially considered and termed a cell adhesion molecule, although it has only weak cell-adhesive properties [1,2]. EpCAM acts as a multi-functional transmembrane protein involved in the regulation of cell adhesion, proliferation, migration, stemness, and epithelial-to-mesenchymal transition (EMT) of normal and neoplastic epithelial cells (summarized in [3,4]). Inter- and intracellular EpCAM signaling mechanisms involve the generation of functionally active fragments that are shed to the extra- and intracellular space. EpCAM and its fragments interact with various proteins including claudins, CD44, and E-cadherin, and regulate growths relevant proteins such as c-Myc, Cyclin A, E, and D1 (summarized in [3,4]).

EpCAM expression has been investigated in a broad range of epithelial neoplasms and its expression levels were found to have a prognostic impact in many tumor types (summarized in [3,5]). Due to its membranous expression in epithelial tumors, EpCAM represents a candidate for targeted cancer therapies and various EpCAM directed monoclonal antibodies have been investigated in clinical phase I/II studies (summarized in [3]). In diagnostic surgical pathology, EpCAM immunohistochemistry (IHC) is employed for the distinction of malignant mesothelioma from primary lung cancer (summarized in [6]) and—due to its broad expression in epithelial neoplasms—as a surrogate pan-epithelial marker for the detection of circulating tumor cells (summarized in [7]).

Although more than 1000 studies have investigated EpCAM immunostaining at least in common tumor types (PubMed, 12 September 2023), EpCAM has not or has only rarely been investigated in many other tumor entities. Furthermore, the results of the various studies are often inconsistent, especially for the most commonly studied tumor types. For example, the reported positivity ranges between 0% and 100% in hepatocellular carcinoma [8,9], lobular breast cancers [8,10], squamous cell carcinoma of the esophagus ([10,11], and of the cervix [10,11]], 0–54% in epithelioid mesothelioma [12,13,14], 17–100% in breast cancers of no special type [11,15], or 51–100% in adenocarcinoma of the lung [16,17,18]. Obviously, technical aspects such as different antibodies and staining protocols as well as different scoring criteria have contributed to these differences, making a comprehensive and standardized analysis of EpCAM expression in different human tumor types highly desirable.

Accordingly, the principle aim of our study was to evaluate EpCAM in human tumors to better comprehend the range of diagnostic applications of EpCAM IHC. For this purpose, more than 14,000 tissue samples from 120 different tumor types and subtypes and 76 non-neoplastic tissues were evaluated for EpCAM protein expression by IHC in a tissue microarray (TMA) format. The same tissue cohort has been studied for immunohistochemical expression of trophoblast cell surface glycoprotein 2 (TROP2) and pan-cytokeratin antibodies (CKpan) before [19,20]. TROP2, also known as tumor-associated calcium signal transducer 2 (TACSTD2), is another druggable membrane glycoprotein which is closely related to EpCAM and has also been shown to occur in many epithelial tumors but not in malignant mesotheliomas (summarized in [19,21]). CKpan markers represent the gold standard for the detection of epithelial cells [22]. In the light of the existing data, it also appeared attractive to us to systematically compare the staining patterns of EpCAM, TROP2, and CKpan antibodies.

## 2. Materials and Methods

### 2.1. Tissue Microarrays (TMAs)

The normal tissue TMA contained 8 samples from 8 different donors for each of 76 different normal tissue types (608 samples on one slide). The cancer TMAs included 14,832 tumor samples from 120 tumor types and subtypes distributed across 52 TMA slides. All cancer samples were obtained from primary tumors. Histopathological data on grade, pT or pN status, and molecular data on HER2, the progesterone receptor (PR), and the estrogen receptor (ER) were available from subsets of breast (*n* = 1475), urothelial (1073), and renal carcinomas (*n* = 1157), as well as of 902 squamous cell carcinomas of different sites of origin. Clinical follow-up data were available from 877 breast, 254 bladder, and 850 kidney cancer patients with a median follow-up time of 43/14/39 months (range 1–88; 1–77; 1–250 months). The composition of the TMAs is described in detail in the Results Section. From previous studies, data on CKpan were available for 13,501 and for TROP2 in 16,024 cases [19,20]. All samples were from the archives of the Institutes of Pathology, University Hospital of Hamburg, Germany, the Institute of Pathology, Clinical Center Osnabrueck, Germany, and the Department of Pathology, Academic Hospital Fuerth, Germany, collected between 2005 and 2020. Tissues were fixed in 4% buffered formalin and then embedded in paraffin. TMA manufacturing was described earlier in detail [23,24]. In brief, one tissue spot (diameter: 0.6 mm) was transmitted from a cancer-containing donor block in an empty recipient paraffin block. The use of archived remnants of diagnostic tissues for the manufacturing of TMAs and their analysis for research purposes as well as patient data analysis without informed patient consent is covered by local laws (HmbKHG, §12) and by the local ethics committee (Ethics Commission Hamburg, WF-049/09). All work has been carried out in compliance with the Helsinki Declaration.

### 2.2. Immunohistochemistry (IHC)

Freshly prepared TMA sections were immunostained on one day in one experiment. Slides were deparaffinized with xylol, rehydrated through a graded alcohol series, and exposed to heat-induced antigen retrieval for 5 min in an autoclave at 121 °C in pH 7.8 Tris-EDTA buffer. Endogenous peroxidase activity was blocked with Dako Peroxidase Blocking Solution™ (Agilent, Santa Clara, CA, USA; #52023) for 10 min. Primary antibody specific against the EpCAM protein (rabbit recombinant, MSVA-326R, MS Validated Antibodies, Hamburg, Germany, 2315-326R, epitope not disclosed by the manufacturer) was applied at 37 °C for 60 min at a dilution of 1:150. Bound antibody was then visualized using the EnVision Kit™ (Agilent, Santa Clara, CA, USA; #K5007) according to the manufacturer’s directions. The sections were counterstained with haemalaun. Staining was predominantly membranous and sometimes accompanied by cytoplasmic positivity. For normal tissue analysis, the staining intensity in the different cell types was recorded as negative (0, no staining), weak (1+ staining intensity), moderate (2+ staining intensity), or strong (3+ staining intensity). Scoring of the staining in tumor tissues was performed according to a standard procedure [25]. In brief, raw data were collected including the percentage of EpCAM-positive tumor cells (estimated) and the staining intensity in a semi-quantitative scale semi-quantitatively (0, 1+, 2+, 3+). The raw data were then used to define the staining results in four groups as follows: negative: no staining at all; weak staining: staining intensity of 1+ in ≤70% or staining intensity of 2+ in ≤30% of tumor cells; moderate staining: staining intensity of 1+ in >70%, staining intensity of 2+ in >30% but in ≤70% or staining intensity of 3+ in ≤30% of tumor cells; strong staining: staining intensity of 2+ in >70% or staining intensity of 3+ in >30% of tumor cells. For the purpose of antibody validation, the normal tissue TMA was also analyzed for EpCAM expression by using the monoclonal mouse anti-EpCAM antibody BER-Ep4 (Agilent, Santa Clara, CA, USA; #IR637, ready to use, pH 6.0, epitope not disclosed by the manufacturer) in the Autostainer Link 48 (Agilent, Santa Clara, CA, USA) according to the manufacturer’s directions. Examples of tumors with different scores are shown in Appendix A.

### 2.3. Statistics

Statistical calculations were performed with JMP 16 software (SAS Insitute Inc., NC, USA). Contingency tables and the chi^2^-test were performed to search for associations between EpCAM immunostaining and tumor phenotype of selected tumor types/subtypes. Survival curves were calculated according to Kaplan–Meier. The log-rank test was applied to detect significant differences between groups. A *p*-value of ≤0.05 was considered as statistically significant.

## 3. Results

### 3.1. Technical Issues

A total of 12,780 (86.2%) of 14,832 tumor samples and at least 4 tissue samples per normal tissue category were interpretable in our TMA analysis. Non-interpretable samples demonstrated a lack of unequivocal tumor cells or an absence of tissue in the respective TMA spots.

### 3.2. EpCAM in Normal Tissues

Using MSVA-326R, EpCAM immunostaining was found to be strong in all epithelial cells of the gastrointestinal tract (strongest staining of neuroendocrine cells; lowest staining of parietal cells of the stomach, where staining was limited to the basolateral membranes), all epithelial cells of the gallbladder and bile ducts, pancreas, salivary glands, Brunner glands, prostate, seminal vesicle, epididymis, respiratory epithelium (lowest staining in the basal cell layers), lung, endocervix, endometrium, fallopian tube, thyroid, parathyroid, and the adenohypophysis. In squamous epithelium, EpCAM staining was generally weak and not always seen. If present, EpCAM was most strongly expressed in the basal cell layers but expression also expanded to the upper third in some samples. Some squamous epithelia only showed few scattered EpCAM-positive cells of the upper cell layers. In the skin, EpCAM was expressed in peripheral germinative cells of sebaceous glands, eccrine glands, and in root sheaths of hair follicles but not in the epidermis. Scattered squamous epithelial cells in tonsil crypts showed a strong staining. Most thymus epithelial cells including corpuscles of Hassall’s showed a weak to moderate EpCAM positivity. Urothelium stained strongly, but sometimes less intensely in umbrella cells. In the kidney, strong staining was seen in the distal tubuli, while staining was less intense and focused to the basolateral membranes in the proximal tubuli. Scattered epithelial cells lining the Bowman capsule were also positive. In the breast glands, luminal cells were strongly positive but myoepithelial cells showed much less or absent staining. In the testis, spermatogonia and spermatocytes stained moderately or strongly but Sertoli and Leydig cells were negative. In the adrenal gland, a faint to moderate membranous staining of adrenocortical cells was occasionally seen. In the placenta, staining was weak to moderate in the cytotrophoblast, faint in the amnion, and moderate in chorion cells. Representative images are shown in Figure 1, and all results of the normal tissues’ analysis are listed in Appendix A. EpCAM staining was absent in hepatocytes, aorta and other blood vessels, muscle, fat, lymph node, spleen, neurohypophysis, and the brain. All cell types found to be EpCAM-positive by MSVA-326R were also positive by using Ber-EP4, although the Ber-EP4 staining was found to be considerably weaker (Appendix A).

EpCAM in cancer. A predominantly membranous EpCAM immunostaining was detectable in 10,513 (82.3%) of the 12,780 analyzable tumors, including 1010 (7.9%) with weak, 1993 (15.6%) with moderate, and 7510 (58.8%) with strong immunostaining. Overall, 99 (82.5%) of 120 tumor categories showed detectable EpCAM expression and 85 (70.8%) tumor categories contained at least one case with strong positivity (Table 1). Among 78 epithelial tumor categories, the EpCAM positivity rate was ≥99% in 46 (59.0%) and 90.0% to 98.9% in 14 categories (17.9%). Tumor categories with particularly high positivity rates included adenocarcinomas and neuroendocrine neoplasms (including small cell carcinomas), as well as germ cell tumors. The total rate of positivity and the fraction of strongly positive tumors tended to be lower in urothelial neoplasms and in squamous cell carcinomas. Among epithelial neoplasms, the lowest rates of positive and strongly positive cases occurred in hepatocellular carcinomas, adrenocortical tumors, renal cell neoplasms, and in carcinomas with a particularly poor differentiation such as sarcomatoid urothelial carcinomas and anaplastic carcinomas of the thyroid. EpCAM positivity was seen in 16.7% of 24 epithelioid and in 30.2% of 53 biphasic malignant mesotheliomas where tumor cell staining was always limited to epithelioid cells. In contrast, EpCAM positivity was seen in 100% of 147 adenocarcinomas of the lung. It is of note that some EpCAM positivity could also be observed in several mesenchymal and hematological neoplasms. Representative images of EpCAM-positive tumors are shown in Figure 2. A ranking order of EpCAM-positive cancers is given in Figure 3. The relationships between EpCAM staining and histopathological or molecular features in breast, urothelial, and renal tumors, and squamous cell carcinomas of different sites are shown in Table 2 and Appendix A. High EpCAM staining was linked to high grade (*p* < 0.0001), distant metastasis (*p* = 0.0006), ER/PR loss (*p* < 0.0001 each), and HER2 positivity (*p* = 0.0009) in breast cancer of no special type, as well as to high grade and to HPV infection in squamous cell carcinomas of different sites (*p* < 0.0001 each; Table 2 and Appendix A). Low EpCAM staining was linked to invasive disease in urothelial carcinoma (*p* < 0.0001) and to high grade in clear cell renal cell carcinomas (*p* < 0.05). Significant associations with overall survival and recurrence free survival were not seen in breast, urothelial, and clear cell renal carcinomas (Appendix A).

EpCAM vs. CKpan in cancer. Data on both EpCAM and CKpan immunostaining were available for 11,053 tumors from 101 tumor categories. Their comparison resulted in a generally high concordance, especially in adenocarcinomas where both EpCAM and CKpan were positive in almost 100% of cases (Appendix A). The positivity rate was higher for EpCAM (100%) than for CKpan (23.2%; *p* < 0.0001) in testicular seminomas and—less markedly—in several neuroendocrine neoplasms. The positivity rate was lower for EpCAM in mesotheliomas, hepatocellular carcinomas, very poorly differentiated tumors (anaplastic thyroid cancer, sarcomatoid urothelial cancer), and—to a lesser extent—in squamous cell carcinomas. A positivity for both markers was very rarely seen in hematologic or mesenchymal tumors.

EpCAM vs. TROP2 (EpCAM2) in cancer. Both EpCAM and TROP2 immunostaining data were available for 10,998 tumors from 101 tumor categories. Although many epithelial neoplasms stained with both antibodies, the fraction of positive cases was often lower for TROP2 than for EpCAM (Appendix A). Squamous cell carcinomas of different sites of origin made up for most tumor categories with a higher rate of TROP2 positivity than seen for EpCAM. The rate of TROP2 positivity was markedly lower than for EpCAM in testicular germ cell tumors, neuroendocrine neoplasms, renal cell tumors, and also in many gastrointestinal adenocarcinomas. In hepatocellular carcinomas, 15.4% were positive for EpCAM alone, 15.4% were positive for TROP2 alone, and only 2.6% for both EpCAM and TROP2. In epithelioid mesotheliomas, 9.1% were positive for EpCAM alone, 13.6% were positive for TROP2 alone, and only 4.5% for both EpCAM and TROP2. A compilation of EpCAM, TROP2, and CKpan is given in Appendix A.

## 4. Discussion

Our standardized analysis of 14,832 tumors provides a comprehensive overview on EpCAM immunostaining in malignant tumors. That the graphical representation of the frequencies of EpCAM expression among different tumor categories resulted in an S-shaped curve reflects the intense EpCAM immunostaining in the vast majority of epithelial neoplasms. The validity of our data is supported by the fact that non-epithelial tumors were most commonly EpCAM-negative as described before [26].

The comparison with CKpan revealed a generally high concordance with EpCAM staining, especially in adenocarcinomas where both markers were positive in almost 100% of cases. This justifies the widespread use of anti-EpCAM antibodies for the detection of circulating tumor cells in these tumor categories (summarized in [7]). The rate of EpCAM positivity was especially lower than of CKpan staining in hepatocellular carcinomas, mesotheliomas, squamous cell carcinomas, and in very poorly differentiated tumors such as anaplastic cancers of the thyroid or sarcomatoid urothelial carcinomas. The particularly low rate of EpCAM positivity in mesotheliomas (26%), hepatocellular carcinomas (14.3%), and squamous cell carcinomas of various sites of origin (59.0–98.3%, mean 84.0%) is in the lower range of previous studies describing EpCAM positivity on average in 32% of hepatocellular carcinomas, and somewhat higher compared to previous studies describing EpCAM positivity on average in 10% of mesotheliomas and 66% of squamous cell carcinomas (Appendix A). EpCAM has therefore been suggested as a tool for the distinction of hepatocellular carcinomas and malignant mesotheliomas from their differential diagnoses [11]. The lower positivity rate of EpCAM as compared to CKpan in squamous cell carcinomas obviously reflects the rather low level of EpCAM expression in normal squamous epithelium, while CKpan staining is usually strong in these tissues [20]. That the rate of EpCAM positivity was somewhat lower than that of CKpan positivity in very poorly differentiated epithelial tumors may be reflective of the higher likelihood of a tumor cell losing the expression of just one protein (EpCAM) as compared to simultaneously abandoning the expression of multiple different intermediate filaments in case of cellular dedifferentiation accompanying tumor development and progression.

The markedly higher rate of EpCAM staining as compared to CKpan in seminoma is consistent with data from Schönberger et al. [27] describing EpCAM positivity in nearly all of 32 analyzed malignant germ cell neoplasms including 7 seminomas. Based on these findings, EpCAM is under evaluation as a therapeutic target in germ cell tumors [28] and has been used to identify circulating germ cell tumor cells in the blood [29]. That EpCAM positivity was more common in several neuroendocrine neoplasms than CKpan positivity fits well with the particularly high EpCAM expression in normal neuroendocrine cells in the gastrointestinal tract. The specific role of EpCAM in normal and neoplastic neuroendocrine cells is unclear, although Cives et al. [30] recently provided evidence for a role of EpCAM in EMT in neuroendocrine tumors.

The comparison with TROP2 revealed that EpCAM is less commonly expressed in squamous cell carcinomas but more abundant in most other epithelial neoplasms especially in germ cell tumors, neuroendocrine neoplasms, renal cell tumors, and in gastrointestinal adenocarcinomas. These findings mirror the findings in normal tissues where—in contrast to what is seen for EpCAM—TROP2 positivity is rare in the normal gastrointestinal tract, absent in the testis, inconspicuous in neuroendocrine cells, and absent in the proximal tubuli of the kidney, while high levels of TROP2 are common in squamous epithelium [19]. The lower prevalence of TROP2 expression in normal and neoplastic epithelial cells shows that TROP2 may not replace EpCAM as a surrogate epithelial marker. However, the comparable but independent staining pattern of both markers in malignant mesothelioma, adenocarcinoma, and squamous cell carcinoma of the lung may suggest that TROP2 may complement the list of biomarkers that are most commonly used for this purpose such as calretinin, WT1, D2-40, cytokeratin 5/6, TTF-1, CEA, EpCAM, and Napsin A [31].

The evaluation of EpCAM expression in the context of clinicopathological and molecular features demonstrated that—depending on the tumor type—both upregulation and reduced expression of EpCAM can be associated with unfavorable tumor features and tumor progression. The strong link between poor differentiation and high EpCAM expression in squamous cell carcinomas from different sites of origin may suggest a role of EpCAM upregulation for the progression of these tumors. In line with these findings, other authors have previously described associations between elevated EpCAM levels and a poor prognosis in squamous cell carcinomas of the head and neck [32,33], the glottis [33], and the esophagus [34,35]. The significant association of high EpCAM levels with positive HPV status argues for a dependence of the EpCAM function and expression on the status of specific molecular pathways in cancer cells. The HPV E6/E7 oncoproteins affect numerous intracellular signaling pathways such as ERK, JAK, YY1, API, NF-kB, AKT, and WNT [36], whose key proteins (including E-cadherin, EGFR, c-Myc, and cyclins A, E, and D1) interact with EpCAM [3,4]. In breast cancer, elevated EpCAM expression was also tightly linked to key molecular features such as the loss of ER and PR expression as well as HER2 overexpression in our study. Soysal et al. described similar findings in a cohort of 1365 breast cancers and found that the prognostic impact of EpCAM expression depended on the molecular subtype [37]. The significant associations between low EpCAM expression and high tumor grade in clear cell renal cell carcinoma and invasive tumor growth in urothelial carcinoma demonstrates that tumor progression can also go along with EpCAM downregulation. Comparable results have previously been described by several authors both in clear cell renal cell carcinomas [38,39,40,41] and in urothelial neoplasms [42]. It is of note that other authors had described reciprocal results and found associations between high-level EpCAM expression and tumor progression in renal cell [43] or between low EpCAM expression and tumor progression in squamous cell carcinomas [44]. These discrepant results are reflective of highly variable results described for the prevalence of EpCAM expression ranging from <20% to >80% in many cancer categories (literature summarized in Figure 4). These discrepancies are likely to be due to the use of different antibodies, staining protocols, and criteria for defining positive cancers.

Considering the large scale of our project, emphasis was placed on a thorough validation of our EpCAM assay. The International Working Group for Antibody Validation (IWGAV) has proposed that antibody validation for IHC on formalin-fixed tissues should include either a comparison of the findings obtained by two different independent antibodies or a comparison with expression data obtained by another independent method [45]. To ensure as much as possible that any antibody cross-reactivity would be detected, 76 different normal tissue categories were employed for our validation experiment. The specificity of our EpCAM assay was supported by the good concordance of MSVA-326R staining with RNA data obtained from three independent RNA screening studies [46,47,48,49]. MSVA-326R staining was almost exclusively seen in organs with documented RNA expression and its staining intensity was the strongest in tissues with the highest RNA levels such as the intestine, thyroid, and parathyroid glands and low in the adrenal gland, a tissue with particularly low EpCAM RNA expression. EpCAM staining by MSVA-326R staining in tissues without documented EpCAM RNA expression such as the liver, placenta, and the thymus was confirmed by comparison with the independent EpCAM antibody Ber-EP4 (Agilent, #IR637, RTU). This comparison also confirmed all specific EpCAM-positive cell types identified by MSVA-326R although the staining intensity obtained by the Ber-EP4 RTU resulted in a considerably weaker staining (Appendix A). We assume that these differences in staining intensity are due to the too-low sensitivity of our Ber-EP4 assay, because MSVA-326R but not BER-EP4 resulted in a significant staining of proximal tubuli in the kidney. At least a moderate intensity of basolateral EpCAM staining in the majority of proximal tubules cells has been suggested as the optimal positive tissue control by NordiQC, a major organization devoted to quality control in IHC (https://www.nordiqc.org/epitope.php?id=44 accessed on 10 October 2023).

## 5. Conclusions

In summary, our data provide a comprehensive overview on EpCAM expression in cancer and confirm the suitability of EpCAM as a surrogate epithelial marker for adenocarcinomas and its diagnostic utility for the distinction of malignant mesotheliomas from pulmonary adenocarcinoma. They also show that—dependent on the tumor type—both upregulation and downregulation of EpCAM can be related to cancer progression. The comparison with CKpan revealed that EpCAM staining is particularly common in seminomas and in neuroendocrine neoplasms, while TROP2 is more commonly expressed in squamous cell carcinomas.

## Figures and Tables

**Figure 1 diagnostics-14-01044-f001:**
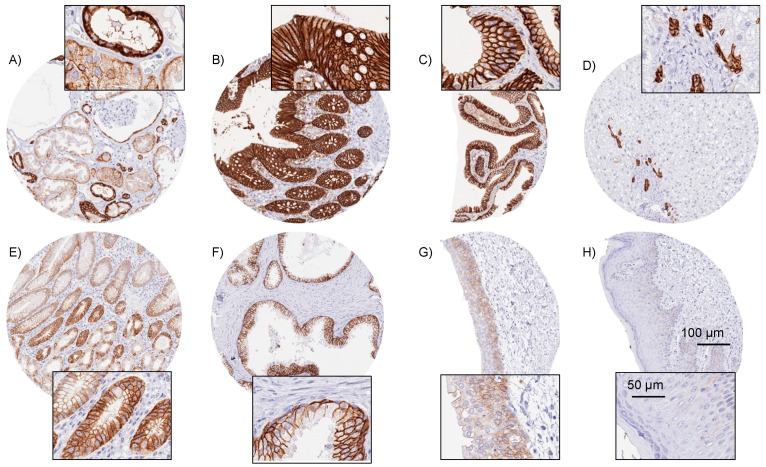
EpCAM immunostaining of normal tissues. In the kidney, EpCAM staining was strong in distal tubuli, moderate in the visceral layer of the Bowman capsule, and weak to moderate and predominantly basolateral in proximal tubuli (**A**). A strong predominantly membranous EpCAM staining occurred in epithelial cells of the colorectum (**B**) and the gallbladder (**C**), as well as in the small bile ducts of the liver (**D**). In the stomach, EpCAM staining was the most intense in neck cells as well as in scattered small (neuroendocrine) cells within corpus and antrum glands (**E**). EpCAM staining was moderate to strong in prostatic epithelial cells (**F**) and weak to moderate in the basal cell layers of the urothelium (**G**). EpCAM staining was absent in the squamous epithelial cells of the skin (**H**). All images at 200× magnification. Insets show details at 400× magnification.

**Figure 2 diagnostics-14-01044-f002:**
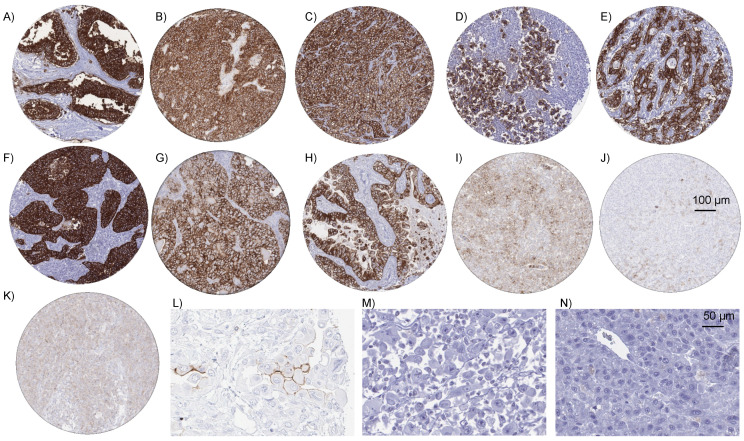
EpCAM immunostaining in cancer. The panels show a strong predominantly membranous EpCAM staining in an adenocarcinoma of the colon (**A**), in an endometrioid carcinoma of ovary (**B**), in a Gleason 5 + 5 = 10 adenocarcinoma of the prostate (**C**), in a seminoma of the testis (**D**), a cholangiocellular carcinoma of the liver (**E**), a small cell neuroendocrine carcinoma of the lung (**F**), in a Warthin tumor of the salivary gland (**G**), and in an adenocarcinoma of the lung (**H**). Occasional weak to moderate membranous staining was found in individual cases of diffuse large cell B-cell lymphoma (**I**), follicular B-cell lymphoma (**J**), angio-immunoblastic T-cell lymphoma (K), and in a rare example of a pleural malignant (epithelioid) mesothelioma with weak to moderate EpCAM staining of a subset of tumor cells (**L**). EpCAM staining was lacking in most malignant (epithelioid) mesotheliomas of the pleura (**M**) and in hepatocellular carcinomas in the liver (**N**). All images are at 200× magnification except (**L**,**M**)) which are at 400× magnification.

**Figure 3 diagnostics-14-01044-f003:**
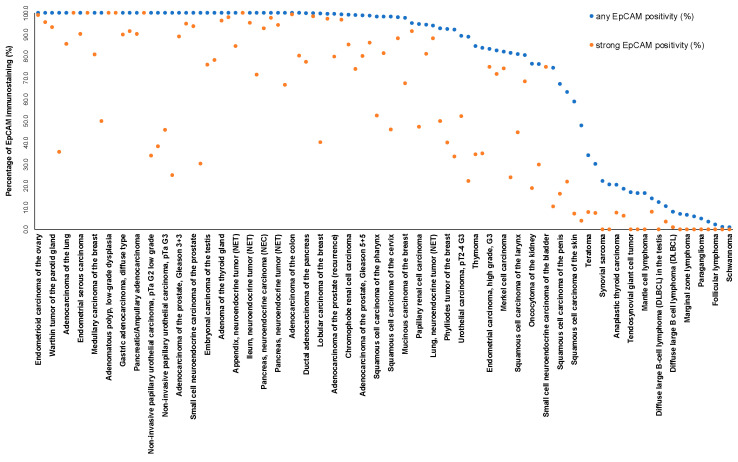
Ranking order of EpCAM immunostaining in cancers. Both the percentage of positive cases (blue dots) and the percentage of strongly positive cases (orange dots) are shown.

**Figure 4 diagnostics-14-01044-f004:**
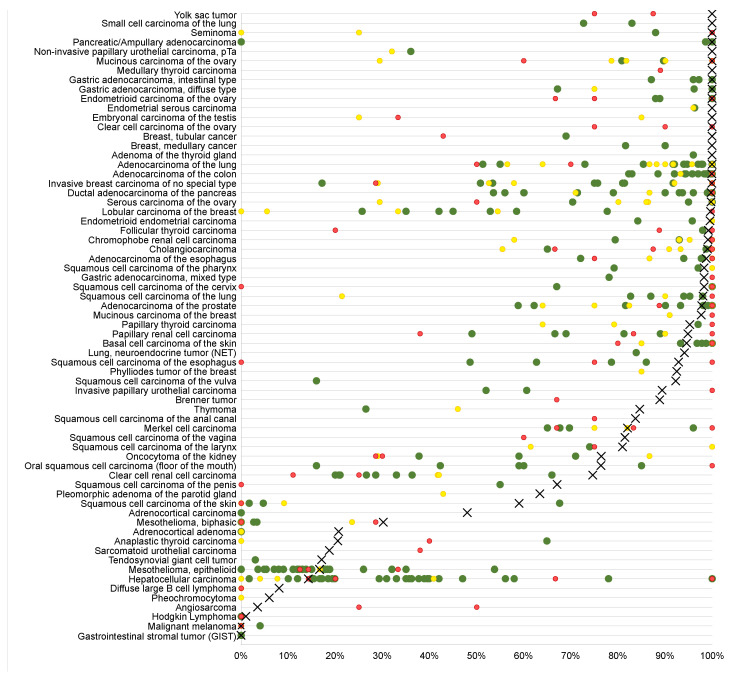
Comparison with EpCAM in previous literature. An “X“ indicates the fraction of EpCAM-positive cancers in the present study, dots indicate the reported frequencies from the literature for comparison: red dots mark studies with ≤10 analyzed tumors, yellow dots mark studies with 11 to 25 analyzed tumors, and green dots mark studies with >25 analyzed tumors.

**Table 1 diagnostics-14-01044-t001:** EpCAM immunostaining in human tumors.

			EpCAM Immunostaining Result				
	Tumor Entity	on TMA (n)	Int. (n)	Neg. (%)	Weak (%)	Mod. (%)	Strong (%)
Tumors of the skin	Pilomatricoma	35	28	89.3	7.1	0.0	3.6
	Basal cell carcinoma of the skin	88	74	5.4	1.4	12.2	81.1
	Benign nevus	29	23	100.0	0.0	0.0	0.0
	Squamous cell carcinoma of the skin	90	83	41.0	41.0	10.8	7.2
	Malignant melanoma	46	41	100.0	0.0	0.0	0.0
	Merkel cell carcinoma	46	39	17.9	2.6	5.1	74.4
Tumors of the head and neck	Squamous cell carcinoma of the larynx	110	105	19.0	23.8	12.4	44.8
	Squamous cell carcinoma of the pharynx	60	59	1.7	22.0	23.7	52.5
	Oral squamous cell carcinoma (floor of the mouth)	130	127	23.6	26.8	19.7	29.9
	Pleomorphic adenoma of the parotid gland	50	41	36.6	19.5	22.0	22.0
	Warthin tumor of the parotid gland	49	45	0.0	0.0	6.7	93.3
	Basal cell adenoma of the salivary gland	15	14	0.0	7.1	57.1	35.7
Tumors of the lung, pleura and thymus	Adenocarcinoma of the lung	196	147	0.0	0.7	13.6	85.7
	Squamous cell carcinoma of the lung	80	51	2.0	2.0	7.8	88.2
	Small cell carcinoma of the lung	16	13	0.0	0.0	0.0	100.0
	Mesothelioma, epithelioid	39	24	83.3	12.5	4.2	0.0
	Mesothelioma, biphasic	76	53	69.8	17.0	5.7	7.5
	Thymoma	29	26	15.4	19.2	30.8	34.6
	Lung, neuroendocrine tumor (NET)	19	17	5.9	0.0	5.9	88.2
Tumors of the female genital tract	Squamous cell carcinoma of the vagina	78	54	18.5	24.1	33.3	24.1
	Squamous cell carcinoma of the vulva	130	116	7.8	34.5	24.1	33.6
	Squamous cell carcinoma of the cervix	129	117	1.7	23.9	28.2	46.2
	Endometrioid endometrial carcinoma	236	219	0.5	0.5	1.8	97.3
	Endometrial serous carcinoma	82	61	0.0	4.9	4.9	90.2
	Carcinosarcoma of the uterus	48	46	17.4	4.3	6.5	71.7
	Endometrial carcinoma, high grade, G3	13	12	16.7	8.3	0.0	75.0
	Endometrial clear cell carcinoma	8	7	0.0	0.0	0.0	100.0
	Endometrioid carcinoma of the ovary	110	96	0.0	0.0	1.0	99.0
	Serous carcinoma of the ovary	559	441	0.2	0.2	1.1	98.4
	Mucinous carcinoma of the ovary	96	70	0.0	1.4	2.9	95.7
	Clear cell carcinoma of the ovary	50	43	0.0	0.0	2.3	97.7
	Carcinosarcoma of the ovary	47	41	19.5	7.3	4.9	68.3
	Brenner tumor	9	9	11.1	33.3	33.3	22.2
Tumors of the breast	Invasive breast carcinoma of no special type	1345	1260	0.1	1.1	18.6	80.2
	Lobular carcinoma of the breast	293	269	0.4	3.7	55.8	40.1
	Medullary carcinoma of the breast	26	26	0.0	0.0	19.2	80.8
	Tubular carcinoma of the breast	27	20	0.0	0.0	50.0	50.0
	Mucinous carcinoma of the breast	58	43	2.3	4.7	25.6	67.4
	Phylloides tumor of the breast	50	40	7.5	2.5	50.0	40.0
Tumors of the digestive system	Adenomatous polyp, low-grade dysplasia	50	46	0.0	0.0	0.0	100.0
	Adenomatous polyp, high-grade dysplasia	50	47	0.0	0.0	0.0	100.0
	Adenocarcinoma of the colon	1882	1499	0.1	0.1	0.5	99.3
	Gastric adenocarcinoma, diffuse type	176	160	0.0	0.0	10.0	90.0
	Gastric adenocarcinoma, intestinal type	174	165	0.0	1.2	7.3	91.5
	Gastric adenocarcinoma, mixed type	62	59	1.7	0.0	16.9	81.4
	Adenocarcinoma of the esophagus	83	80	1.3	3.8	8.8	86.3
	Squamous cell carcinoma of the esophagus	75	70	7.1	27.1	15.7	50.0
	Squamous cell carcinoma of the anal canal	89	80	16.3	22.5	26.3	35.0
	Cholangiocarcinoma	113	104	1.0	6.7	18.3	74.0
	Hepatocellular carcinoma	50	49	85.7	6.1	0.0	8.2
	Ductal adenocarcinoma of the pancreas	612	459	0.2	1.5	20.9	77.3
	Pancreatic/Ampullary adenocarcinoma	89	72	0.0	2.8	6.9	90.3
	Acinar cell carcinoma of the pancreas	16	15	0.0	0.0	0.0	100.0
	Gastrointestinal stromal tumor (GIST)	50	48	100.0	0.0	0.0	0.0
	Appendix, neuroendocrine tumor (NET)	22	13	0.0	0.0	15.4	84.6
	Colorectal, neuroendocrine tumor (NET)	12	8	0.0	0.0	0.0	100.0
	Ileum, neuroendocrine tumor (NET)	49	44	0.0	0.0	4.5	95.5
	Pancreas, neuroendocrine tumor (NET)	97	89	0.0	0.0	5.6	94.4
	Colorectal, neuroendocrine carcinoma (NEC)	12	7	0.0	0.0	28.6	71.4
	Gallbladder, neuroendocrine carcinoma (NEC)	4	3	0.0	0.0	33.3	66.7
	Pancreas, neuroendocrine carcinoma (NEC)	14	14	0.0	7.1	0.0	92.9
Tumors of the urinary system	Non-invasive papillary urothel. ca., pTa G2 low grade	177	138	0.0	18.8	47.1	34.1
	Non-invasive papillary urothel. ca., pTa G2 high grade	141	112	0.0	16.1	45.5	38.4
	Non-invasive papillary urothelial carcinoma, pTa G3	187	161	0.0	15.5	38.5	46.0
	Urothelial carcinoma, pT2-4 G3	623	518	10.6	10.4	26.6	52.3
	Small cell neuroendocrine carcinoma of the bladder	20	20	25.0	0.0	0.0	75.0
	Sarcomatoid urothelial carcinoma	25	16	81.3	12.5	0.0	6.3
	Clear cell renal cell carcinoma	857	793	25.3	34.6	29.5	10.6
	Papillary renal cell carcinoma	255	232	5.2	18.1	29.3	47.4
	Clear cell (tubulo) papillary renal cell carcinoma	21	20	0.0	25.0	50.0	25.0
	Chromophobe renal cell carcinoma	131	116	0.9	3.4	10.3	85.3
	Oncocytoma of the kidney	177	153	23.5	37.9	19.6	19.0
Tumors of the male genital organs	Adenocarcinoma of the prostate, Gleason 3 + 3	83	83	0.0	0.0	10.8	89.2
	Adenocarcinoma of the prostate, Gleason 4 + 4	80	80	0.0	0.0	5.0	95.0
	Adenocarcinoma of the prostate, Gleason 5 + 5	85	85	1.2	0.0	18.8	80.0
	Adenocarcinoma of the prostate (recurrence)	258	193	0.5	1.0	18.7	79.8
	Small cell neuroendocrine carcinoma of the prostate	19	16	0.0	6.3	0.0	93.8
	Seminoma	621	594	0.0	15.0	54.7	30.3
	Embryonal carcinoma of the testis	50	25	0.0	12.0	12.0	76.0
	Yolk sac tumor	50	32	0.0	12.5	9.4	78.1
	Teratoma	50	38	65.8	13.2	13.2	7.9
	Squamous cell carcinoma of the penis	80	79	32.9	36.7	13.9	16.5
Tumors of endocrine organs	Adenoma of the thyroid gland	114	109	0.0	2.8	0.9	96.3
	Papillary thyroid carcinoma	392	294	4.8	0.0	3.7	91.5
	Follicular thyroid carcinoma	154	128	0.8	0.8	1.6	96.9
	Medullary thyroid carcinoma	111	100	0.0	0.0	2.0	98.0
	Anaplastic thyroid carcinoma	45	39	79.5	10.3	2.6	7.7
	Adrenal cortical adenoma	50	29	79.3	17.2	3.4	0.0
	Adrenal cortical carcinoma	26	25	52.0	40.0	4.0	4.0
	Pheochromocytoma	50	50	94.0	2.0	4.0	0.0
Tumors of haemato-poetic and lymphoid tissues	Hodgkin lymphoma	103	98	99.0	0.0	1.0	0.0
	Small lymphocytic lymphoma, B-cell type	50	50	100.0	0.0	0.0	0.0
	Diffuse large B cell lymphoma (DLBCL)	114	112	92.0	1.8	5.4	0.9
	Follicular lymphoma	88	88	97.7	1.1	1.1	0.0
	T-cell non-Hodgkin lymphoma	24	24	100.0	0.0	0.0	0.0
	Mantle cell lymphoma	18	18	83.3	16.7	0.0	0.0
	Marginal zone lymphoma	16	15	93.3	6.7	0.0	0.0
	Diffuse large B-cell lymphoma (DLBCL) in the testis	16	16	87.5	12.5	0.0	0.0
	Burkitt lymphoma	5	4	100.0	0.0	0.0	0.0
Tumors of soft tissue and bone	Tendosynovial giant cell tumor	45	41	82.9	17.1	0.0	0.0
	Granular cell tumor	53	36	100.0	0.0	0.0	0.0
	Leiomyoma	50	49	100.0	0.0	0.0	0.0
	Leiomyosarcoma	87	83	100.0	0.0	0.0	0.0
	Liposarcoma	132	121	100.0	0.0	0.0	0.0
	Malignant peripheral nerve sheath tumor (MPNST)	13	12	100.0	0.0	0.0	0.0
	Myofibrosarcoma	26	26	100.0	0.0	0.0	0.0
	Angiosarcoma	73	57	96.5	1.8	1.8	0.0
	Angiomyolipoma	91	75	100.0	0.0	0.0	0.0
	Dermatofibrosarcoma protuberans	21	15	100.0	0.0	0.0	0.0
	Ganglioneuroma	14	14	92.9	7.1	0.0	0.0
	Kaposi sarcoma	8	4	100.0	0.0	0.0	0.0
	Neurofibroma	117	110	100.0	0.0	0.0	0.0
	Sarcoma, not otherwise specified (NOS)	74	69	100.0	0.0	0.0	0.0
	Paraganglioma	41	41	95.1	2.4	2.4	0.0
	Ewing sarcoma	23	14	100.0	0.0	0.0	0.0
	Rhabdomyosarcoma	6	5	100.0	0.0	0.0	0.0
	Schwannoma	121	115	99.1	0.9	0.0	0.0
	Synovial sarcoma	12	9	77.8	11.1	11.1	0.0
	Osteosarcoma	43	31	100.0	0.0	0.0	0.0
	Chondrosarcoma	38	19	100.0	0.0	0.0	0.0

Abbreviation: Int.: interpretable, Neg.: negative, Mod.: moderate.

**Table 2 diagnostics-14-01044-t002:** EpCAM immunostaining and cancer phenotype.

			EpCAM Immunostaining Result				
		n	Negative (%)	Weak (%)	Moderate (%)	Strong (%)	*p*
Invasive breast carcinoma of no special type	pT1	613	0.0	1.0	20.4	78.6	0.2200
	pT2	435	0.0	0.7	15.6	83.7	
	pT3-4	89	0.0	0.0	18.0	82.0	
	G1	184	0.0	1.6	28.3	70.1	<0.0001
	G2	597	0.0	1.2	20.8	78.1	
	G3	396	0.0	0.0	9.3	90.7	
	pN0	505	0.0	1.0	21.2	77.8	0.4295
	pN+	366	0.0	0.8	17.8	81.4	
	pM0	206	0.0	0.5	23.3	76.2	0.0006
	pM1	109	0.0	0.0	7.3	92.7	
	HER2 negative	880	0.0	0.8	19.1	80.1	0.0009
	HER2 positive	121	0.0	0.8	6.6	92.6	
	ER negative	212	0.0	0.9	5.7	93.4	<0.0001
	ER positive	741	0.0	0.5	19.8	79.6	
	PR negative	402	0.0	0.2	10.9	88.8	<0.0001
	PR positive	592	0.0	1.0	21.8	77.2	
	non-triple negative	781	0.0	0.6	18.7	80.7	0.0001
	triple negative	141	0.0	0.7	5.7	93.6	
Clear cell renal cell carcinoma	ISUP 1	247	26.7	40.5	26.7	6.1	0.0010
	ISUP 2	237	21.9	31.2	33.8	13.1	
	ISUP 3	209	23.9	32.5	29.2	14.4	
	ISUP 4	46	45.7	17.4	30.4	6.5	
	Fuhrman 1	41	24.4	31.7	34.1	9.8	0.0578
	Fuhrman 2	440	24.8	36.4	29.5	9.3	
	Fuhrman 3	213	23.0	31.5	30.5	15.0	
	Fuhrman 4	54	42.6	20.4	31.5	5.6	
	Thoenes 1	282	25.2	40.4	26.2	8.2	0.0005
	Thoenes 2	395	22.8	30.1	33.7	13.4	
	Thoenes 3	71	42.3	25.4	26.8	5.6	
	UICC 1	346	25.1	34.4	29.8	10.7	0.1599
	UICC 2	38	13.2	36.8	36.8	13.2	
	UICC 3	92	18.5	29.3	34.8	17.4	
	UICC 4	73	31.5	28.8	34.2	5.5	
	pT1	451	25.1	34.4	30.2	10.4	0.9123
	pT2	77	27.3	36.4	24.7	11.7	
	pT3-4	215	25.6	31.2	32.6	10.7	
	pN0	126	26.2	29.4	31.7	12.7	0.6486
	pN+	19	31.6	26.3	21.1	21.1	
	pM0	114	21.9	33.3	29.8	14.9	0.0465
	pM+	74	36.5	27.0	31.1	5.4	
Urothelial bladder carcinoma	pTa G2 low	138	0.0	18.8	47.1	34.1	<0.0001
	pTa G2 high	112	0.0	16.1	45.5	38.4	
	pTa G3	163	0.0	15.5	38.5	46.0	
	pT2-4	486	11.3	10.1	27.4	51.2	
	pTa G2 low	138	1.4	17.4	47.1	34.1	<0.0001
	pTa G2 high	112	0.9	15.2	45.5	38.4	
	pTa G3	163	1.8	14.7	38.0	45.4	
	pT2	139	12.9	9.4	24.5	53.2	
	pT3	225	9.3	12.0	27.1	51.6	
	pT4	106	12.3	6.6	32.1	49.1	
	pT2	139	12.9	9.4	24.5	53.2	0.5484 *
	pT3	225	9.3	12.0	27.1	51.6	
	pT4	106	12.3	6.6	32.1	49.1	
	G2	23	8.7	8.7	30.4	52.2	0.9686 *
	G3	447	11.2	10.1	27.3	51.5	
	pN0	275	11.6	10.5	30.9	46.9	0.0582 *
	pN+	170	8.4	9.6	22.2	59.9	
Squamous cell carcinomas of different sites **	pT1	266	12.4	25.9	20.3	41.4	0.2135
	pT2	239	13.0	29.3	20.5	37.2	
	pT3	136	11.0	27.2	19.9	41.9	
	pT4	128	14.1	17.2	15.6	53.1	
	pN0	310	12.6	22.3	21.0	44.2	0.2559
	pN+	307	8.5	26.4	18.9	46.3	
	G1	25	48.0	24.0	8.0	20.0	<0.0001
	G2	377	15.6	33.4	17.5	33.4	
	G3	257	7.8	19.5	26.8	45.9	

* only in pT2-4 urothelial bladder carcinomas, ** oral, pharynx, larynx, esophagus, cervix, vagina, vulva, penis, anal, and lung; abbreviation: pT: pathological tumor stage, G: grade, pN: pathological lymph node status, pM: pathological status of distant metastasis, PR: progesterone receptor, ER: estrogen receptor, ISUP: International Society of Urological Pathology, UICC: Union for International Cancer Control, Fuhrman: Grading according to G.A. Fuhrman, Thoenes: Grading according to w. Thoenes.

## Data Availability

The raw data supporting the conclusions of this article will be made available by the authors on request.

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
