# Peer review of "Epithelial Cell Adhesion Molecule (EpCAM) Expression in Human Tumors: A Comparison with Pan-Cytokeratin and TROP2 in 14,832 Tumors"

_diagnostics, 2024, doi:10.3390/diagnostics14101044_

Round 1
Reviewer 1 Report
Comments and Suggestions for Authors
The main question addressed by the research is epithelial cell adhesion molecule (EpCAM) expression in human tumors (a comparison with pan-cytokeratin and TROP2 in more then 14 000 tumors. This paper is solid and original with a huge cohort assessed. This survey is incredibly important because there is some gaps for many tumors to fill for increase diagnostic accuracy. Other publications cover only one marker and involved much less participants. From my point of view the authors perform a comprehensive investigation with TMA to assess as many different tumors from different localisations as possible. The results of this investigation proved that EpCAM can be used as a surrogate epithelial marker for adenocarcinomas and also has a diagnostic utility for the distinction of malignant mesotheliomas. In addition, the authors demonstrated that EpCAM can stain in seminomas and in neuroendocrine neoplasms more common than CKpan and TROP2 which could be also implemented in routine practice. The conclusion is totally in agreement with the survey results. The cited references are appropriate and cutting-edge. So I recommend this article to be published.
Author Response
Hamburg, 26.04.2024
Dear Editor,
In reply to the comments of reviewer 1, we are grateful to the reviewer for his positive assessment of our work, and the time devoted to our manuscript.
With kind regards,
Ronald Simon
Original reviewer comment: The main question addressed by the research is epithelial cell adhesion molecule (EpCAM) expression in human tumors (a comparison with pan-cytokeratin and TROP2 in more then 14 000 tumors. This paper is solid and original with a huge cohort assessed. This survey is incredibly important because there is some gaps for many tumors to fill for increase diagnostic accuracy. Other publications cover only one marker and involved much less participants. From my point of view the authors perform a comprehensive investigation with TMA to assess as many different tumors from different localisations as possible. The results of this investigation proved that EpCAM can be used as a surrogate epithelial marker for adenocarcinomas and also has a diagnostic utility for the distinction of malignant mesotheliomas. In addition, the authors demonstrated that EpCAM can stain in seminomas and in neuroendocrine neoplasms more common than CKpan and TROP2 which could be also implemented in routine practice. The conclusion is totally in agreement with the survey results. The cited references are appropriate and cutting-edge. So I recommend this article to be published.

Reviewer 2 Report
Comments and Suggestions for Authors
The manuscript titled 'Epithelial cell adhesion molecule (EpCAM) expression in hu- 2 man tumors: A comparison with pan-cytokeratin and TROP2 in 3 14,832 tumors' will be a significant contribution towards tumour pathology. It is important to study the expression of EpCAM in different tumours and immunohistochemistry on tissue microarrays is one of the most appropriate methods. Here authors have used normal TMAs and 14,832 tumour samples, which is quite impressive. IHC was validated by two antibodies to EpCAM. Overall, the manuscript is quite interesting and content is significant.
Please see the comments below.
1) Although the discussion is in detail, it would be worthwhile to conclude the study either as the last paragraph of discussion or a separate section.
2) Please add the future directions/perspectives of this study.
3) Please explain or expand the figure legends for Supplementary figs 3-6.
4) There are quite a number of articles on EpCAM expression in TMAs of different tumour types. Please include them in the discussion. For eg:
Fong et.al., J.Clin Pathol, 2014
Sanchez et.al, Eur J Histochem 2022
5) Only a few references were recent (2021&2022). Please add more recent references from 2021 to 2024.
6) EpCAM is reported to be a cancer stem cell marker in different tumour types. Have you considered this and tried to clinically correlate the EpCAM expression levels in tumours to their treatment type such as Primary surgery or Neo-adjuvant chemotherapy ?
7) It would be worthwhile to add a supplementary table (Log rank test results) with correleation data of EpCAM expression in each tumour types and subtypes with their overall survival or recurrence free survival.
8) Have you analyzed the prognostic significance of the combined expression (both positive) of EpCAM and CKpan/ TROP2 to overall survival or progression free survival for each tumour type?
9) Please include when the tumour samples and normal tissues were collected and what was the median follow-up of these patients? Will the increase in the follow-up duration change the correlation of EpCAM expression with OS or RFS?
10) Please see the attached PDF for more comments.

Author Response
Hamburg, 26.04.2024
Dear Editor,
In reply to the comments and suggestions of reviewer 2, we made the following changes:
1) Although the discussion is in detail, it would be worthwhile to conclude the study either as the last paragraph of discussion or a separate section.
Reply: We have now added our conclusion to the last section., page 16, lines 415-422.
2) Please add the future directions/perspectives of this study.
Reply: The reviewer suggests adding the future directions. However, as per instructions for authors, the “future directions” section is for review type publications only.
3) Please explain or expand the figure legends for Supplementary figs 3-6.
Reply: Following the reviewer’s request, we have substantially expanded the legends to Suppl. Figs. 3-6 on page 16-17.
4) + 5) There are quite a number of articles on EpCAM expression in TMAs of different tumour types. Please include them in the discussion. For eg: Fong et.al., J.Clin Pathol, 2014, Sanchez et.al, Eur J Histochem 2022. 5) Only a few references were recent (2021&2022). Please add more recent references from 2021 to 2024.
Reply to 4-5): The reviewer suggests adding specific and additional references. We have now added more and recent references to Suppl. Table 2 and in Figure 4 in the discussion.
6) EpCAM is reported to be a cancer stem cell marker in different tumour types. Have you considered this and tried to clinically correlate the EpCAM expression levels in tumours to their treatment type such as Primary surgery or Neo-adjuvant chemotherapy ?
Reply: Although this would be of interest, we cannot perform such analysis because we do not have data on patient therapies.
7) It would be worthwhile to add a supplementary table (Log rank test results) with correleation data of EpCAM expression in each tumour types and subtypes with their overall survival or recurrence free survival.
Reply: Survival data were available from urinary bladder cancers, renal cell carcinomas, and breast cancers of no special type. The Log rank test result p-values are shown in Supplementary Figure 3.
8) Have you analyzed the prognostic significance of the combined expression (both positive) of EpCAM and CKpan/ TROP2 to overall survival or progression free survival for each tumour type?
Reply: We calculated these associations but there were no significant results (breast NST: OS for EpCAM+Ckpan: p=0.6872, OS for EpCAM+TROP2: p=0.0967; bladder cancer pT2-4: OS for EpCAM+Ckpan: p=0.1472, OS for EpCAM+TROP2: p=0.5606; renal cell carcinoma: OS for EpCAM+Ckpan: p=0.3769, OS for EpCAM+TROP2: p=0.5512; renal cell carcinoma: PFS for EpCAM+Ckpan: p=0.1769, OS for EpCAM+TROP2: p=0.9913.
9) Please include when the tumour samples and normal tissues were collected and what was the median follow-up of these patients? Will the increase in the follow-up duration change the correlation of EpCAM expression with OS or RFS?
Reply: We have now added the time when the samples were collected in the materials and methods section, page 2, line 93-94. There is no scheduled update of the follow-up data.
10) Please see the attached PDF for more comments.
Reply: We made the following changes in reply to the comments made in the pdf:
- Detailed that the tumor samples were distributed across 52 TMA slides (page 2, line 82)
- Explained that the epitopes of both antibodies have not been disclosed (page 3 lines 109 and 127)
- Mentioned that staining was predominently membranous (page 3 lines 113-114)
- Modified Figures 1 and 2 and their legends according to the suggestions of the reviewers (added scalebars, added additional tumor types, showed selected tissues at higher magnification)
- Corrected the typos on page 5 line 225 and in the footnote of table 2
- Explained Fuhman and Thoennes in the footnote of table 2
- Adjusted the font size of Figure 4
We thank the reviewer for his valuable comments and suggestions, and the time devoted to our manuscript.
With kind regards,
Ronald Simon
